# The Big and the Great: A Reconstruction of Zhuangzi's Philosophy on Transcendence

## Limei Jiang

Center of Value & Culture, School of Philosophy, Beijing Normal University, Beijing 100875, China; jianglimei@bnu.edu.cn

**Abstract:** This essay attempts to demonstrate the logic of Zhuangzi in his different attitudes toward "debate on big and small" by bringing into discussion the two versions of translation in the English languages, which provide a new approach to analyze the difference in the controversial commentaries on Zhuangzi. This essay points out that the ideal of "free and easy wandering" is a type of positive pleasure. By means of rational thinking and imagination, one's searching for the external values is turned into the internal pursuit for wisdom in the transformation of things. Zhuangzi's theory of transcendence not only provides the subject with multi-perspectives, but also substitutes the self-identity with self-value. Through the interaction between self-awareness and self-reaction, the subject can be unified with the great Dao through purposive activities. However, Guo Xiang's commentary cancels the necessity of self-cultivation and negates the purposefulness of the subject, which underestimates the value of Zhuangzi's construction of transcendence.

**Keywords:** positive pleasure; purposiveness; self-awareness; self-reflection; no-self

---

## 1. Introduction

The "debate on big and small" (*xiaoda zhi bian*小大之辨) was raised in the first chapter, *Free and Easy Wandering* (*xiaoyaoyou*逍遥游), of the *Zhuangzi*. It is also discussed in other chapters such as *On Making Things Fit Together* (*qiwulun*齐物论) and *Autumn Waters* (*qiushui*秋水). However, there are two different kinds of attitudes towards the big and the small in the text, which are: (1) it praises the big and disparages the small; (2) the coming together in unity of the big and the small. The former has been regarded as an expression of a philosophy of transcendence and the latter has been viewed as relativism or skepticism which rejects distinctions. The transmitted commentaries have also revolved around these topics resulting in controversy. Guoxiang郭象 argues for the consistency of the status that the large and small can both achieve, while Lin Xiyi林希逸, Luo Miandao罗勉道, Shi Deqing释德清, and Wang Fuzhi王夫之 hold that the large is much better than the small. In order to solve this confliction, Liu Xiaogan put forth a two-world framework of Zhuangzi: an earthy or secular world and transcendent and spiritual world (Liu 2014, p. 204). Chad Hansen explains the reason and logic in Zhuangzi's relativist position from a pluralistic and naturalistic perspective on language (Hansen 1992), while Coutinho uses theoretical appreciation to point out that Zhuangzi is a pragmatist and not a relativist (Coutinho 2004, p. 165).

This essay argues that Zhuangzi's philosophy is a transcendent philosophy with humanism. By utilizing two different English translations of "*da*大", this essay attempts to demonstrate the internal logic of Zhuangzi's theory on the relationship between the two and also explores the possibility of connecting the *Free and Easy Wandering* chapter with the *On Making Things Fit Together* chapter. Moreover, this essay also points out some problems and insufficiencies with Guo Xiang's interpretation of *Zhuangzi*. By offering a close reading of selected parts of Zhuangzi with respect to the topic of big

and great, the essay aims to demonstrate Zhuangzi's theory of individual transcendence from the perspectives of value/fact, self-transformation/transformation of things, and self-identity/self-value.

## 2. The Big and the Great

In Chinese, the concept of "big/great" (*da*大) and "small" (*xiao*小) has many connotations and a high degree of ambiguity. They are used not only to describe the size of things, but also to describe the quantity and height of things. In addition, "big" and "small" in Chinese can be used as an adjective and as a noun; however, when used as a noun it is mainly used as word type.[1] In contrast, there are two types of translation in English translations of the *Zhuangzi*, and in essays which discuss the "debate on big/great and small". One translation is "big and small," used for example by Watson (1968), and Ziporyn (2009). Another kind of translation is "great and small," employed for example by A.C. Graham (Graham 1981), James Legge (Legge 1891), Mair (Mair 1994), Hyun Höchsmann with Yang Guorong (Höchsmann and Yang 2007) and Liu Xiaogan (Liu 2014), among others. Chad Hansen simultaneously uses both of these translations; he points out that stories in the first chapter deal with the great and small, while large and small in the second chapter discuss the relativity of comparative distinctions (Hansen 1992, p. 272). If we look at how these translators treat instances of "*da*大" in the Free and Easy Wandering chapter (the form underneath), we may conclude that Graham, Ziporyn, Höchsmann and others do not specifically differentiate "*da*," while Legge and Mair both use "*da*" very precisely. Both of them tend to use "big" or "huge" to describe the largeness of a thing's shape, and choose "great" to explain formless and imageless things, emotions, knowledge, etc.

| Translator | Great | Big | Large | Huge |
|---|---|---|---|---|
| Graham | Wings, tree | Boat, bird's back | | |
| Legge | Wind, knowledge, accumulation of water, words | | Boat, calabash, bottle-gourds, tree, yak | |
| Mair | Knowledge, speech, fear | Fish (kun), tree | Gourd | Bird (Peng) |
| Watson | Wings | Gourd, tree | Boat | Kun, gourd |
| Ziporyn | Gourd | Gourd | Gourd | Kun, tree |
| Höchsmann and Yang | Wings | Yak, clouds | Boat, bowl calabash, tree, yak | |
| Liu | | Peng, bird, gourd | | Bird, gourd, Peng, |

In English, what "big" and "small" point to is the differences of a things' quantity, degree, shape in space, while "great" denotes a state of superiority in addition to its original meaning of "large." However, in Chinese, these two concepts are used in combination. If we bring the difference between the English words "big" and "great" into our discussion, we may have a different approach to demonstrate Zhuangzi's attitude toward "*da*大," which will also provide a different perspective for understanding divergences in later commentaries on the "debate of big and small."

Zhuangzi prefers to employing very concrete beings to provide inspirations for imaginative space, which helps to create a psychological feeling of being "vast and far off." The *Free and Easy Wandering* chapter opens with the back of Kun and the wings of Peng, orientating readers towards the imagination of their largeness. Zhuangzi then describes even greater spaces through proceeding descriptions: (1)

---

[1]  Chris Fraser makes a distinction between word type and word occurrence in the using of nouns in Chinese. He states that a word type is a word as identified generally, apart from its use in any particular context, while a word occurrence is a particular instance of a word in a particular sentence (Fraser 2007).

the Northern Darkness and Southern Darkness which can contain Kun and Peng and (2) the movement from the Northern Darkness to the vaster cosmic space of the Southern Darkness. The size of boats, winds, trees-of-heaven, and gourds in the following paragraphs are all used to express "largeness" in terms of their spatiality. He Bo of the *Autumn Waters* chapter and the story of Ren Gongzi fishing in the *Externalizing Things* (*waiwu*外物) chapter also express an appreciation for "largeness," in order to illustrate the shallow knowledge of the subject due to their shortsighted views and limited mind. The quest for the essence of "*da*" not only fits with the rules of life and growth, but also reflects people's pursuit for space.[2] Confucius also mentioned that "benevolent persons enjoy mountains, wise persons enjoy rivers," people become more open and carefree in their hearts when they enjoy the vastness of mountains and oceans, obtaining an appreciation for beauty and spiritual happiness. Moreover, Zhuangzi uses his exasperation with the turtledove and cicada to compare the difference in abilities and in the relative superiority of opinions that come from the size of the body in order to explain the exquisite spirit contained in "pursuing to fly south" (*tunan zhi zhi*图南之志). He uses the term "its vision below" (*shixia* 视下) to explain the diverse viewpoints people have according to their respective positions. Based on the foundation of the original spatial meaning of "*da*," Zhuangzi expands it to temporal and spiritual levels. He states that "small knowledge does not come up to large knowledge, and that few years do not come up to many years," "big/many and small/few" (both use *da*大and *xiao*小) being used here portrays the quantity of one's years and of one's wisdom. Through the discussion of the relativity of the big/great and the small (especially the relativity of the big), Zhuangzi rebukes the people's blind opinion of themselves as high, reminding them of the limitations inherent in the shape of their bodies, lives, and wisdom and thus he "gives us a greater and more accurate appreciation of our true place in the world." (Ivanhoe 1996, p. 210) As a result, people are encouraged to break out of their own limitations to seek the unlimited, constantly seeking for the big and bigger until they finally become great.

Zhuangzi artfully uses the continuity between "big" and "great" to turn people's search for greatness into a subjective intuitional feeling and rationally deduced autonomy, people's quest for an externally great space is been turned into a quest for an internal wisdom. Therefore, in Zhuangzi's opinion, obtaining spiritual freedom is the process of a subject moving towards "dependencelessness" (*wudai*无待). He calls this "free and easy (*xiaoyao*逍遥)," which deserves to be called positive pleasure. However, other chapters in the *Zhuangzi* contain another attitude towards the big and the small. In the *On Making Things Fit Together* chapter, Zhuangzi points out that: "There is nothing under heaven larger than the tip of a downy hair at the end of autumn, but Mount Tai is small. There is no greater longevity than that of a child who dies in infancy, but Peng Zu died young." Zhuangzi elaborates the discussion of big and small to talk about the limitlessness of time and space from the view of the great cosmos. On the one hand, it is very general that the sizes of things are all relative because it originates in the comparison between objects. From the perspective of the limitless, "viewing things in their differences, see what is large in what is large [for one thing], thus there will not be a single thing which is not large; see what is small in what is small [for one thing], thus there will not be a single thing which is not small. Know that Heaven and Earth are grains of rice and that the tip of downy hair in autumn is a mountain," the big is small and the small can also be big, that's why Zhuangzi concludes that "a blade of grass and a tree trunk, Li and Xishi, things strange and unfamiliar, *dao* pervades them all." On the other hand, all the things are the same in essence because of the component qi. Zhuangzi holds that the size of things is no more than the result of the movement of the condensation and dispersal of *qi*; therefore, the small envying the large or the large oppressing the small are meaningless in terms of the essence of things. In the *Primacy of Cultivating Life* (*Yangshengzhu*养生主) chapter, Zhuangzi's argument "to seek what is without end with what has end is dangerous" expresses his rejection to

---

[2] Chen Shaoming holds that the tendency to emphasize largeness and de-emphasize smallness reflects people's necessity of space. People pursuit for broader space for their activities of the body and their subsistence and growth, which is prominent in the time of hunter-gatherers when good physiques and power are highly appreciated (Chen 2018).

seek an absolute and unconfined existence with one's limited and confined body, life, and wisdom. Zhuangzi's employment of the fish Kun also has the meaning of equalizing the large and the small.[3] According to *Erya—Shiyu* ("explaining fish"): "Kun means fish eggs." In general, the eggs of fish are called *kun*鲲. The *Luyu*鲁语 says that "catching young fish and fish eggs are prohibited" (*yu jin kuner*鱼禁鲲鲕), Wei Zhao comments "*kun* means fish eggs." Duan Yucai says that "before fish have been born they are called *kun*, *kun* is the same as *ruan*卵 (also 'egg')."[4] All these classics treat *kun* as fish eggs or yet-to-be-born fish. According to these interpretations, Zhuangzi uses a sarcastic method to say that the largest is also the smallest (the tiny fish egg), Kun's transformation into Peng becomes a metaphor for something small turning into something very big.

As a representative commentary, Guo Xiang offers a different opinion on the transformation. When Guo Xiang said: "Now, as for the differences of large and small, as long as a thing is in its place proper, then that thing will take up its own nature, function with its own capacities, each thing will obtain its own portion, and be equally free and easy. How could triumph and defeat be permissible therein?" According to his logic, he acknowledges that all things have their differences in terms of their objective existence, this is a discussion of big and small. However, when he goes a step further arguing that the unity that both big and small can achieve is the same, he actually denies the difference in transcendent capabilities that come from the quantity in size, longevity, and wisdom of things. Moreover, after Guo Xiang equalizes the big/great and the small, then the "constraints of envy and desire" that come from "the small envying the large" and the "large desiring the small" are avoided through eliminating the possible restraint and loss of spiritual freedom. Compared to Guo Xiang, the argument of "fulfilling one's nature and be free and easy" of Zhi Daolin支道林 can be understood as the process of each person fully realizing their own greatness, but he still admits that there exists a difference in the greatness which each person can realize.

However, in the theoretical structure of the "praising greatness school" (*chongdapai*崇大派) formed by scholars such as Lin Xiyi, Wang Fuzhi, Lin Yunming, Zhang Taiyan章太炎 and others, pay great attention to Zhuangzi's own appraisal of "enlarge and transform it" (*da er hua zhi*大而化之). They provide proofs that the sarcasm revealed in the *Zhuangzi*'s story of the cicada and the turtledove laughing at greatness with their smallness expresses Zhuangzi's appreciation of a lofty and vast realm, which is in accordance with his artfully technique used in understanding "great" based on "big". The "praising greatness school" is already reflected in later school of Zhuangzian thought. In the entire book of the *Zhuangzi*, there are the terms "great man" (*daren*大人) and "utmost man" (*zhiren*至人),[5] which are used to describe the ideal human state of being. However, if we strictly investigate the text, we will discover that the inner chapters only use the term "utmost man" and there are no signs of "great man (*daren* 大人)." But when it comes to the outer and miscellaneous chapters, other than "utmost man," "spirit-like man" (*shenren*神人), and the "sagely-minded man" (*shengren*圣人), there is also the "great man." The Autumn Waters chapter brings up the saying that "the great man is without self" which is quite similar to the *Free and Easy Wandering* chapter's saying that "the utmost man is without self". They substitute the "great man" with the "utmost man", subconsciously expressing that only through the way of being big and being great that one can get the unity of *dao*. The *Xuwugui*徐无鬼chapter also gives the attributes of "being without merit" (*wugong*无功) and "being without reputation" (*wuming*无名) to the "great man," consciously using the term "great man" to bring these three personalities

---

[3] There is a controversy in comprehension on the meaning of the character "*kun*鲲." Lu Deming(陆德明), Cui Zhuan(崔) and Wang Fuzhi all maintain that it should be understood as "large fish" (*dayu*大鱼). Lu Deming says: "Kun鲲is pronounced "kun昆" in Xu, it is the name of a large fish." (Lu 1980, p. 225) The *Yupian—Yubu*玉篇·鱼部 ("jade chapter—fish section") also says: "Gun鲧is a large fish." (Sibu Congkan四部丛刊.经部) Cui Zhuan says: "Kun鲲is a whale." (Guo 2014, p. 3) Wang Fuzhi also thinks that "as a fish it is large, it is also large when it becomes a bird. Even though it transforms, its largeness does not change. The amount of largeness is fixed." (Wang 2009, p. 75).

[4] Cited by Guo Qinfan (Guo 2014). *Zhuangzi Jishi* 庄子集释. Beijing, Zhonghua Shuju. p. 3.

[5] The "utmost man" cited here is Graham's translation of "*zhiren* 至人". As for other translations there are: "perfect man" (Legge), "ultimate man" (Mair), "consummate man" (Ziporyn), and "complete man" (Höchsmann and Yang 2007).

under the command of one. However, later commentators have brought up a different attitude in the discussion around the "debate on big and small." For example, Fang Yizhi方以智 in the Ming Dynasty thought that we cannot treat Kun and Peng as an analogy of Zhuangzi himself, the attitude of viewing the big/great as beautiful or unifying the big/great and the small are both kinds of prejudices. Fang broke through the difference between big/great and small and raised a new concept of "the middle" (*zhong*中). He cited the *Zhongyong's*中庸saying that "use the middle when in between two extremes" (*liangduan yongzhong*两端用中) to point out that the utmost person is capable of being without prejudices, not being obstructed by their current perspectives ([Fang 2016](), p. 103). When he is dealing with the outside world, he is able to achieve the ideal status "non-arousal of the emotions of delight, anger, sorrow, and happiness" (*xinu aile zhi weifa*喜怒哀乐之未发) and act according to the way of the middle.

Therefore, when we return again to the discussion on the "debate on big and small," we will discover that Zhuangzi did not deny that there is a factual difference in the "big" and the "small" of things, What he opposed to is directly relating value to the fact. His discussion on the relativity of "big" and "small" aims to obtain a comprehensive grasp of things as they truly are by breaking through the prejudiced likes and dislikes of people. Even if he connects "big" with "great," he does not deny the possibility of transforming from the "small" to the "great," he emphasized that regardless of a thing's size, both have the possibility of obtaining absolute and transcendent *dao*, in another word, each of us can take the "great man" as our ideal. People are inspired to break through the limitations from bodies and lifespans, finding their way to personal greatness and make this greatness be one with *dao*.

## 3. Transformation of the Self and the Transformation of the Things

Zhuangzi's discussion on the "debate on big/great and small" may be traced back to Hui Shi. Two of Hui Shi's famous paradoxes being recorded in the last chapter of *Zhuangzi* are discussions on the relationship between big and small:

> The largest unit has nothing outside it, I call it the great unity. The smallest unit has nothing within it, I call it the small unity.

> What has no thickness cannot be piled up, and yet it extends for a thousand miles. Heaven is as low as earth, and the mountain as level as the lake. ([Ziporyn 2009](), p. 124)

Hui Shi brings up the concepts of "the largest" (*zhida*至大) and "the smallest" (*zhixiao*至小) in order to make an argument that the sizes of things are all relative. There are even more similarities between Zhuangzi and Hui Shi in the *On Making Things Fit Together* chapter, but Hui Shi's "largest" stops at the level of formal concepts and he does not provide with any content. Zhuangzi, though, says "Viewing Hui Shi's skills against the Course of Heaven and Earth, they look like the busy labors of a mosquito or a fly." ([Ziporyn 2009](), p. 125) Here, he uses the concept of the "way of Heaven and Earth" to describe the state of the "largest." The "way of Heaven and Earth" used here does not simply mean "big," and has received the great influence of Laozi's philosophy. Laozi argues for the mutual reliance and mutual transformation between the big/great and the small rather than making distinctions (chp. 2). More significantly, Laozi associates the great with *dao*, in the *Daodejing*, he uses the term "great *dao*" (*dadao*大道) four times (chp. 18, 25, 34, 53). Compared with the great *dao*, the ten thousand things are all small because they are bounded to some kind of form. Laozi states that besides the great *dao*, there are three other "greats:" great Heaven, great Earth, and the great king (*daodejing*, chp. 25). The *Shuowen Jiezi* directly quotes these "four greats" to explain the meaning of the character for "great" (*da*大). Wang Bi王弼 said in his commentaries about these four, "as for the greatest, it is only *dao* which is so." (*da zhi ji, qiwei dao hu*大之极其唯道乎). Zhuangzi associates the great with the great *dao*; however, this will cause some kind of problem: is it possible for the limited and confined one to be really unified with the great *dao*? As people's cognition of big and small comes from the comparison of themselves with other objects with a strong subjective consciousness, how can one

cast off the subjectivity to get unification with *dao*? How to transform from "having a self" to having "no-self"?

In Zhuangzi's opinion, the process of transformation relies on the strong ambitions of the subject. In the story of the transformation of Kun into Peng, "pursuing to fly south," "rousing himself in flight," (*nu er fei*怒而飞) "accumulation" (*ji*积) and other words are used to express strong motivations of the fish, which shows that Zhuangzi advocates the consciousness and practice in the process of self-transformation. In the cultivation method of "sitting and forgetting" (*zuowang*坐忘), "letting fall" (*duo*堕) and "casting out" (*chu*黜) are also reflections of willful restraints of the subject. In the 19 years' skillful practicing of the art with knife, Butcher Ding consciously substitutes his senses with the spiritual mind, sheds away the methods of scraping and slicing to let his knife roam where there is open space. Liu Xiaogan addresses this as Zhuangzi's "preparation for transcending" (Liu 2014, p. 199), which adopts an optimistic attitude to reach for transcendent freedom and moreover break through the established results of people's encounters and occasions. However, the purposiveness that Zhuangzi argues for is not utilitarian with the consideration of reputation and benefits. The purposiveness of Zhuangzi consists in an appeal to improve the subject's wisdom in order to achieve true "clarity" (*ming*明) by differentiating the big/great and small of "knowing." Zhuangzi presents an important concept of the "transformation of things" (*wuhua*物化) to offer a solution of the confliction between the "having a self" and "no self" in the process of entering into the natural state.

The "transformation of things" comes from the story of Zhuangzi dreaming he is a butterfly, which also contains the metaphor of the big and small. It says:

> Once Zhuang Zhou dreamt he was a butterfly, fluttering about joyfully just as a butterfly would. He followed his whims exactly as he liked and knew nothing about Zhuang Zhou. Suddenly he awoke, and there he was, the startled Zhuang Zhou in the flesh. He did not know if Zhou had been dreaming he was a butterfly, or if a butterfly was now dreaming it was Zhou. Surely, Zhou and a butterfly count as two distinct identities! Such is what we call the transformation of one thing into another. (Ziporyn 2009, p. 21)

Zhuangzi admits that everything is in the process of change, the subject may be transformed into different species (fish to bird), different types (animals to plants), and different shapes (the liquids turning into the wild fire). This reflects Zhuangzi's abolishment of making distinctions among the ten thousand things. In the story of the dream, Zhuangzi and the butterfly belong to two different subjects, each has its own distinct identity, but the mutual transformation between Zhuangzi and the butterfly is not only the transformation of the small and the large but also includes the transformation of consciousness. When Zhuangzi becomes the butterfly, he is a butterfly fluttering around, and when the butterfly becomes Zhuang Zhou, he is a stiff and rigid Zhuangzi. The interchangeability of subjects is not only expressed in the identity and roles of the subject, but also provides the subject with multiple perspectives to break through the original prejudices produced from its physical and emotional situation. Moreover, the subject can also reflect on its own position from the perspective of the other. Zhuangzi attempts to break down the concept of taking oneself as what is correct through the interchanging of different perspectives, allowing people to return to a state of emptiness so that one can be empty and await things, use the heart-mind like a mirror, and realize the loss of self through the method of "forgetting" (*wang*忘). The satisfaction of Butcher Ding and Zhuangzi flying as a butterfly is the ideal status when one flows in unison with Heaven and Earth and achieves a great personhood with no self at all.

Conversely, Guo Xiang explains "two vermin" with "the Peng bird and the cicada", denying the universal meaning of the will of "flying south" (*tunan*图南). His commentary directly cancels the necessity of self-cultivation by substituting the search for a new state of being with one's own appropriateness (*zishi* 自适). As a result, the value of Zhuangzi's construction of the "free and easy" is underestimated, the "great" loses its significance of superiority. More importantly, Guo Xiang negates the purposefulness of the person united with *dao*, causing people

to stagnate in a state of purposelessness. Individuals not only lose motivation in the process of self-transcendence, but also come to be satisfied with the result of self-contentment and self-transformation. However, "self-transformation" is the natural unfolding of *dao.* In the *Daodejing*, in chapter 57, "self-transformation" is just the result that the people naturally grow and mature that is brought about by the ruler's enacting the governance of non-coercion (*wuwei*无为), expressing a positive mutual correspondence between superiors and inferiors. Compared with Laozi, Zhuangzi's discussions on transformation mainly concern the change of things. The inner chapters of the *Zhuangzi* only contain the "transformation of things", while "self-transformation" only appears in the outer and miscellaneous chapters. Later, when Guo Xiang uses "self-transformation" to interpret "solitary transformation" (*duhua*独化), what he emphasizes is the significance of "self" in the process of self-generativity and "spontaneous becoming" (*ziwei*自为) of things. By negating any transcendent external power, Guo Xiang establishes a solitary individual isolated from others, which makes it impossible to make distinctions between things, not to mention to make comparison of the greatness the individuals achieve.

Compared to Guo Xiang, Zhuangzi constructs a transcendent self by fully expanding the spirituality and imaginative powers of the subject. The text of the *Zhuangzi* in Sima Qian's eyes is full of "nonsense and absurd sayings" (*huangtang miuyou zhi yan*荒唐谬悠之言), the "analogies" and "goblet words" of Zhuangzi's "three kinds of sayings" use fabricated stories and plots to resist common sense, that is rooted in the cultural elements of Chu such as myths and rituals. With imagination, a subject can enter into a fantastic world of his own fabrication and liberate himself from the influence of the mundane concerns of wealth, reputation, and of the emotions. As to the problem of big and small, due to the transformation of things, the ten thousand things are able to exchange bodies, sizes, and lives with others by themselves. The similarity between oneself and others makes it possible for one to change from a specific existence to a being with universality. Through observing the flow of great transformations, one gradually can turn oneself from great into an absolute until one achieves inclusive greatness. Therefore, making distinctions between big and small provides the subject with an opportunity to transcend itself. By getting rid of benefit and reputation, one can change oneself from small to big, through the purposive practice of the subject. While with the recognition that the big/great and the small are equal, the individual is able to transform itself freely, realizing self-transcendence to the great and completing the personality of the great man.

## 4. Self-awareness and Self-reflection

Although Zhuangzi raises the problem of "spirit' combined with the body and mind, he emphasizes the individual's activity of mind more because he considers it as the crucial element in the spiritual movements. In the *On Making Things Fit Together* chapter, Zhuangzi analyzes the concept of "self" (*wo*我) in Nanguo Ziqi's reply of "I lost my self." He illustrates the process of spiritual transcendence as the interaction between self-awareness and self-reflection by the metaphor of wonderful music. Nan Guo says:

> What has happened here is simply that I have lost me. Do you understand? You hear the piping of man but not yet the piping of the earth. You hear the piping of the earth but not yet the piping of the heaven. (Ziporyn 2009, p. 21)

Zhuangzi holds that the movement of "hearing" is not the enjoyment of external sounds, but an internal exploration of the movement of the mind and the *qi*气. Therefore, with the deep investigation of the wills of mind, there is a great development of the cognition of self. Self-awareness is the knowledge of the uniqueness of the self by comparing one's current situation of the body, role, and life with others. These characteristics constitute the consciousness of "I" which establishes a concept of "self-identity". Zhuangzi maintains that "self" is been built up by the continuous reflection on the idea of "I". However, he not only opposes the subjectivity based on external and conventional values, but also attempts to break through the human limitations created from this self-identity.

First, the size of one's body is an important part of self-image, but Zhuangzi argues that lives of people consist of the condensed or dispersed *qi*气and do not possess any stability or universality. Therefore, it cannot be served as the foundation for self-identity. Second, Zhuangzi rejects the identity being defined by one's social group and opposes understanding someone through his social roles or family identity. In the *On Making Things Fit Together* chapter, he refused to regard a person's mundane identity as their unique characteristic in the story "Yao preparing to resign the empire to Xu You," Xu You replied:

> You rule the world, and the world is thus already ruled however you rule it. If I were nonetheless to take your place, would I not be doing so for the name? But name is merely a guest of what is really substantial. Shall I then play the role of the guest? (Ziporyn 2009, p. 6)

As the guest of the reality, the social identity can be replaced by any other subjects with the same qualification, just like emperor Yao's position in the empire, it cannot be regarded as the uniqueness. Besides, in order to get a better reputation and higher positions in the organization, one cannot keep the personal independence and one's own attributes which make one stand out among others. Accordingly, Zhuangzi used a great many characteristics from skill and appearance to describe people's uniqueness, for example, Butcher Ding, Carpenter Shi, Shaman Jixian, Cripple Shu, Toeless Shushan, Hunchbacked Tuo, Hunchback Limpleg the lipless cripple, and others. Therefore, by means of self-reflection, one can get rid of the uniqueness which is grasped through identity, and then one will no longer stay in a world of duality. Compared with Zhuangzi, Guo Xiang emphasizes the concept of "portion" (*fen*分) that comes from self-identity. In this perspective, the search for the transcendence falls back into the self-comfort and contentment in their social duties. Guo's theory of transformation is nothing more than a continual growth and development of the subject based on one's identity, with no transcendent characteristic. However, when Zhuangzi/Guoxiang says "I have lost myself," this implies that the subject will realize that the self and external things are interconnected in the mind and in the body. The realization of the "self" become a completely clear and particular self-conscious flow of life and, moreover, this life is not solitary but establishes a fluid world in its interconnectedness with external things. Nanguo Ziqi's state of being "content as if he has lost his partner" (*dayan si sang qi ou*嗒焉似丧其耦) is just this abandonment of the form of the "self" established through external things and mental activity. This state is not necessarily like Yancheng Ziyou's observation—"a body like withered wood and a heart-mind like dead ashes"—conversely, his inner mind obtains a new recognition through reflection. By replacing self-identity with self-value, the subject succeeds in breaking through prejudices and dynamically moves towards the more transcendent world of *dao* through activities.

Zhuangzi made *dao* the ultimate goal for man's search for value. However, this *dao* is not an abstract absolute concept, instead, it is the bringing of *dao* into the world of things through the process of transcending the self and the revealing and embodiment of the function of *dao* through the establishment of the subject. This is also an important reason why he brought up the "wandering" (*you*游) in "free and easy wandering" (*xiaoyaoyou*逍遥游). There are two different worlds in Zhuangzi's theory, in the world of things, people differentiate the big from the small, take the big for the good or beautiful or laugh about the small. However, when it comes to the world of *dao*, the relative relationships of things are cancelled by the absolutely great; mutually transforming, the small can become the big/great (Kun and Peng) and the big/great can also become the small (Zhuang Zhou and the butterfly). Besides the mutual transformation of form and consciousness, an independent individual is being changed to a self-in-relationships. Through the persistent practice of the subject with purposiveness, the self accomplishes its unification with the great, the world of *dao* and the world of things can be connected with each other, and in this the self is turned into no self.

Berling (1985) has pointed out that Zhuangzi's self has a unique soteriological meaning. We can easily find the similarity between Zhuangzi's transcendent philosophy and religious transcendence. In his well-known statement "Heaven and Earth are born alongside me and the ten thousand things and I are one," there are two approaches implied in it: one consists in losing the self when one is

unified with Heaven, Earth and the ten thousand things, which means that the transcendent power can emerge in the specific being; the other consists in one becoming the sage because there is clearly a distinction between one and others which demonstrates that the transcendence can be embodied by oneself. These two are in accordance with each other in the recognition of "I". On the one hand, Zhuangzi's philosophy discovers the limited nature of humans from the perspective of the spatially and temporally unlimited universe. He says "My life has an end but my knowledge is without end. To follow what is without end with what has an end is dangerous," (Primacy of Cultivating Life养生主). People's aspiration of seeking unlimited knowledge with their limited lives is an approach of "doing study" (*weixue*为学), it can only result in the replica of knowledge. Conversely, reflecting on the meaning of existence through internalizing knowledge, one can stand firm on one's awareness of the subjectivity to obtain "great knowledge" (*dazhi*大知) and become the unique one out of the ten thousand things, The approach of "doing *dao*" (*weidao*为道) is to turn the small existence into a big one with many possibilities so that people can grow into an actual life that is as large as Heaven and Earth and the ten thousand things. However, on the other hand, the personality of the "great self" (*dawo*大我) created through the effort of the subject can also set a principle for the universe. Xu Fuguan said: "Personality is the highest form of the theory. Therefore, the highest art is something that takes the highest personality as its object." (Xu 2010, p. 62) The personalities of the utmost man, spirit-like man, and great man are not established just for themselves, but conversely, the realization of one's self can act as a pronoun for *dao*, in the relationship of self and external things the characteristics and functions of *dao* are opened up, as well as serving as a principle for the values of others.

Zhuangzi outlines a great number of people who have obtained *dao* or become hermits throughout the text; for example, there are Xu You and Nanguo Ziqi mentioned above, as well as Lian Shu, Chang Wuzi, and others. They are trying to guide ordinary people to find their ways of the great Dao in their responses to others. However, rather than deifying them as the religious god, Zhuangzi still regards them as ordinary people who possess some aspects of *dao*, which affirms the human capabilities of inquiring into their own power. As far as the daemonic person on Miaogushe Mountain and the "true people of old" (*Dazongzhi*大宗师) who have some mystical characteristics are concerned, their supernatural powers are just proofs that the individual has been integrated with the great *dao*. It cannot be interpreted to show that Zhuangzi accepts the supernatural power as an essential approach to get transformed. Besides, this kind of spirituality is not something that is gifted by an external God, but something got from the arousal of its power of subjectivity. Therefore, the philosophy of Zhuangzi tries to establish the ideal of the "unity of Heaven and Humaneness" (*tianrenheyi*天人合一) through the transformation of the self, the great *Dao* is not an authority outside of the self, but a principle embodied in and by the self. Zhuangzi believes in the abilities of the human being, trying to liberate themselves from the restriction of politics, knowledge, and space. With the fulfillment of the subjectivity, one becomes a potential power with various possibilities. Therefore, in comparison with religious characteristics, Zhuangzi's description of transcendence has more humanistic characteristics.

## 5. Conclusions

The two different attitudes to the "debate on big/great and small" in the *Free and Easy Wandering* and *On Making Things Fit Together* chapters can be unified. The coming together of large and small in unity and the praise of the high and lofty both aims to help the individual to realize the limits of body, knowledge, and wisdom. Through recognition of self-reflection on self-identity, the foundation of one's own values will no longer be established on the basis of morality, society, or other people. The external quest is turned into an internal reflection and the spiritual transformation will be realized through purposeful actions. Zhuangzi's theory of the transformation of things helps the individual to break through "having a self" and entering into "no self." In this process, the small is changed into something bigger, ultimately it is unified with the great *Dao*. Although Zhuangzi's philosophy has some similarities with the concept of religious transcendence, the self he constructed lacks divinity and the power of the transformation comes from the individual; therefore, it is still a humanistic representation.

**Funding:** This research was funded by THE NATIONAL SOCIAL SCIENCE FUND OF CHINA, grant number 14ZDB003.

**Conflicts of Interest:** The author declares no conflict of interest.

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
