# Peer review of "The Big and the Great: A Reconstruction of Zhuangzi’s Philosophy on Transcendence"

_religions, doi:10.3390/rel10010030_

Round 1
Reviewer 1 Report
Referencing and citation style needs to be made coherent, following the Journal’s guidelines. In-text citations and references should be given also for citations from pre-modern commentaries. Commentators should be named with pinyin and Chinese characters at first occurrence.
I suggest to consult with the editors about the question of citations of premodern sources (I am not familiar enough with the conventions of the Journal; very often premodern sources are handled differently than contemporary or scholarly sources.
There are many problems with the English language, I have edited and corrected much (track changing is on), the mistakes should be corrected and several unclear sentences (noted with comments) need to be reworded.
It is mandatory that the English language problems get addressed before publication.
Apart from the language issues, this is an interesting and relevant article.

Author Response
Referencing and citation style needs to be made coherent, following the Journal’s guidelines. In-text citations and references should be given also for citations from pre-modern commentaries. Commentators should be named with pinyin and Chinese characters at first occurrence.
I have made modification of citation and reference with t the guidelines from the editor and added the pinyin and Chinese character of all the commentators.
I suggest to consult with the editors about the question of citations of premodern sources (I am not familiar enough with the conventions of the Journal; very often premodern sources are handled differently than contemporary or scholarly sources.
There are many problems with the English language, I have edited and corrected much (track changing is on), the mistakes should be corrected and several unclear sentences (noted with comments) need to be reworded.
Thank you for the corrected part in the pdf you sent, and I have got help from native speaker to help me with the language.
Reviewer 2 Report
There are two significant problems with this paper. The first is that the author(s) claim at the beginning of the paper to provide a comparative treatment of Zhuangzi's philosophy by bringing him into conversation with Kant and Buber. What the paper delivers, however, is a close reading of selected commentators on Zhuangzi, with one undeveloped reference to Kant's theory of the sublime and Buber on the experience of transcendence. The author(s) should either develop these comparative references or remove them and revise the paper accordingly. The second is that there are places in the text which are unclear, and hamper the ability of readers to grasp the argument being presented. For example, the author(s) claim at the beginning that they argue that Zhuangzi presents a "transcendent philosophy with humanism." The vagueness of this phrase is frustrating. My suggestion is that the author(s) decide what kind of paper they wish to write (cross-cultural comparative approach or a close reading of selected commentaries) and revise the paper accordingly.
Author Response
There are two significant problems with this paper. The first is that the author(s) claim at the beginning of the paper to provide a comparative treatment of Zhuangzi's philosophy by bringing him into conversation with Kant and Buber. What the paper delivers, however, is a close reading of selected commentators on Zhuangzi, with one undeveloped reference to Kant's theory of the sublime and Buber on the experience of transcendence. The author(s) should either develop these comparative references or remove them and revise the paper accordingly. The second is that there are places in the text which are unclear, and hamper the ability of readers to grasp the argument being presented. For example, the author(s) claim at the beginning that they argue that Zhuangzi presents a "transcendent philosophy with humanism." The vagueness of this phrase is frustrating. My suggestion is that the author(s) decide what kind of paper they wish to write (cross-cultural comparative approach or a close reading of selected commentaries) and revise the paper accordingly.
1.I have followed your advice and delete the parts related to Kant and Buber(including the reference) and will stick on the close reading of the Zhuangzi with respect to the topic of big and great.
2.I have asked got from native speakers with the language.
3.I have deleted the part of “humanism” in the last paragraph, abstract and introduction.
Round 2
Reviewer 2 Report
Authors responded to the criticisms raised in the previous review.